# Comparative Study of Metastasis Suppression Effects of Extracellular Vesicles Derived from Anaplastic Cell Lines, Nanog-Overexpressing Melanoma, and Induced Pluripotent Stem Cells

**DOI:** 10.3390/ijms242417206

**Published:** 2023-12-06

**Authors:** Celine Swee May Khoo, Takuya Henmi, Mikako Saito

**Affiliations:** 1Department of Biotechnology and Life Science, Tokyo University of Agriculture and Technology, 2-24-16, Naka-cho, Koganei, Tokyo 184-8588, Japan; 2Bioresource Laboratories, Tokyo University of Agriculture and Technology, 2-24-16, Naka-cho, Koganei, Tokyo 184-8588, Japan

**Keywords:** melanoma B16-F10, Nanog, iPS cells, anaplastic cells, metastasis

## Abstract

Previous studies have demonstrated that extracellular vesicles (EVs) derived from an anaplastic mouse melanoma cell line made using *Nanog* overexpression of F10 (*Nanog*^+^F10) suppressed the metastasis of *Nanog*^+^F10. Here, an induced pluripotent stem (iPS) cell line was focused as a more anaplastic cell line, potentially producing EVs with higher metastasis-suppressive effects. The EVs were introduced into the tail vein nine times before introducing *Nanog*^+^F10 cells. Two weeks later, the liver and lung were resected and metastatic colonies were quantified. The involvement of macrophages (invasion inhibiting ability, phagocytic activity) and cytotoxic T cells (cytotoxicity) was evaluated using J774.1 and CTLL-2 cell lines. iPS EVs showed similar level effects to *Nanog*^+^F10 EVs in every item relevant to metastasis suppression. Differential expression analysis of miRNAs in EVs and functional network database analysis revealed that dominant regulatory miRNAs were predicted. The candidate hub genes most highly associated with the metastasis suppression mechanism were predicted as six genes, including *Trp53* and *Hif1a*, for *Nanog*^+^F10 EVs and ten genes, including *Ins1* and *Kitl*, for iPS EVs. Regarding the mechanism, *Nanog*^+^F10 EVs and iPS EVs were very different. This suggests synergistic effect when used together as metastasis preventive vaccine.

## 1. Introduction

It is an extremely serious problem that treatment-resistant cancer cells (chemoresistance [1,2] or radiation resistance [3,4]), either existing within the primary cancer or newly generated, may eventually cause metastasis. The number of such cells is extremely small and it is difficult to determine their location within the body. Standard treatments for cancer such as surgery, anticancer drug administration, and radiation therapy are not suitable for this problem due to technical and practical issues. In contrast, vaccines appear to be potentially effective, and extracellular vesicles (EVs) and exosomes have attracted much attention as resources for vaccines.

EVs are secreted from most cells, circulate throughout the body, and are taken up by distant cells. The type of molecular information conveyed by EVs is complex, and much research has been conducted on the interactions between cancer cells and immune cells [5,6,7,8,9]. From the perspective of preventing metastasis, it is desirable that immune cell-derived EVs exhibit a cancer-suppressing effect or that cancer cell-derived EVs stimulate immune cells and enhance their cancer-suppressing effect. However, such functions are limited to the following few cases.

Loading tumor antigens into dendritic cells (DCs) during EV production along with poly(I:C) (a molecular pattern associated with viral infection) resulted in a vaccine capable of inducing strong activation of melanoma-specific CD8+T cells and recruitment of cytotoxic CD8+T cells [10]. When topotecan acted on breast cancer cells, it induced the breast cancer cells to release EVs containing DNA that activated dendritic cells [11]. Vaccination of transgenic HLA-A2/HER2 mice with a single dose of EVs from adenoviral vector-transfected dendritic cells activated the cytolytic function of CD8+T cells against breast cancer cells in vitro and reduced tumor growth in vivo [12]. Melanoma-derived EVs entered DCs and activated the TLR3-TRIF signaling to increase Treg production and tumor growth. At the same time, however, the EVs activated CD300a, which inhibited the TLR3-TRIF signaling, causing a decrease in Treg production and tumor growth [13]. This suggests a dual role of tumor-derived EVs. A murine colorectal carcinoma cell line of CT-26-derived exosomes containing miR-124-3p mimic was applied to tumors in mice and resulted in the inhibition of tumor growth and an increase in median survival times [14]. Other clinical trials were also conducted and summarized in [15] but their therapeutic effects were limited.

This situation may reflect the fact that we still do not fully understand the complex and diverse performance of EVs. Therefore, new ideas are considered important in order to create EVs with significantly higher performance than before. Focusing on anaplastic cells was our new idea. We initially created a melanoma cell line that overexpressed *Nanog* as a malignant cancer cell model. *Nanog* is an anaplastic factor, and a poorly differentiated state is a property of cancer stem cells [16]. The *Nanog*-overexpressing cell line exhibited the same properties as malignant cancer cells with high metastatic potential [17]. However, while EVs derived from the original melanoma cell line, B16-F10 (F10 EVs), showed a metastasis-promoting effect, EVs derived from the *Nanog*-overexpressing melanoma cell line (*Nanog*^+^F10) (*Nanog*^+^F10 EVs) showed a metastasis-suppressive effect [18]. There have already been many reports regarding the correlation between *Nanog* and the malignant transformation of cancer cells [19,20,21]. In 14 out of 15 types of cancer including breast cancer, colon cancer, large intestine cancer, liver cancer, and skin cancer (melanoma), the higher the expression level of *Nanog* (*NANOG*), the higher the malignancy of the cancer cells (in vitro proliferation ability, migration ability, invasion ability, drug resistance, and in vivo metastatic ability). However, this is the first report that EVs secreted from a *Nanog*-overexpressing cancer cell line exhibited a metastasis-suppressive effect, which became a major impact from the perspective of preventing cancer metastasis.

The next challenge was to examine the EVs’ specificity for cancer types and to improve the metastasis-suppressive effect. Regarding specificity for cancer types, it was expected that anaplastic cancer cells produced through *Nanog* overexpression could be applied to other cancer types. In fact, in the case of colon cancer, *Nanog*-overexpressing colon-derived EVs showed the effect of suppressing colon cancer metastasis (unpublished data). On the other hand, with regard to improving the metastasis-suppression effect, we focused on the concept of anaplastic cells as resources for EVs, rather than the concept of malignant cancer cells, and specifically investigated the use of embryonic stem cells (ES cells) and induced pluripotent stem (iPS) cells. Regarding ES cells, it was reported that EVs secreted from them promoted the recovery of injured cells [22,23]. Therefore, they may assist the recovery of cells that have been attacked by cancer cells, and, as a result, the invasion and proliferation of cancer cells may be inhibited. Similar properties were expected for iPS cell-derived EVs (iPS EVs). In this study, we selected iPS cells, which are easier to use than ES cells from a practical standpoint, and aimed to quantitatively analyze the metastasis-suppressing effect of iPS EVs in comparison with the effect of *Nanog*^+^F10 EVs. According to the references [24,25], we conducted differential expression analysis of the miRNA in EVs and functional network database analysis to predict the candidate hub genes for the feedback for our future experimental studies using mice and cells.

## 2. Results

### 2.1. Transfer of EVs to Various Organs

EVs were isolated from F10 cells, *Nanog*^+^F10 cells, and iPS cells using CD81 as a marker (Figure 1A). As a premise for analyzing the effects of EVs, it is necessary to estimate the extent to which EVs introduced through the mouse tail vein migrate to each organ. Therefore, EVs fluorescently labeled with the CellVue™ NIR815 Cell Labeling Kit (Invitrogen, Waltham, MA, USA) were introduced through the tail vein at 5 µg/100 μL.

The fluorescence intensities in the liver, spleen, and bone marrow were higher than their respective controls (Figure 1B). The fluorescence intensity of each organ was highest in the liver, followed by the spleen, but was extremely low in other organs (Figure 1C). In the liver, there was no difference between the uptake amounts of F10 EVs, *Nanog*^+^F10 EVs, and iPS EVs. Similarly, there was no difference in the uptake of *Nanog*^+^F10 EVs and iPS EVs in the spleen. Therefore, even if there is a difference between the efficacies of *Nanog*^+^F10 EVs and iPS EVs, this is not necessarily due to a difference in the level of uptake into organs.

### 2.2. Effects of EVs as a Vaccine

*Nanog*^+^F10 EVs and iPS EVs were derived from anaplastic cells, but *Nanog* expression was contrasting in both EVs. In the case of *Nanog*^+^F10 EVs, *Nanog* was highly expressed [10], but in iPS EVs, the expression was greatly reduced compared to the original iPS cells. In other words, among the eight anaplastic genes, the expression levels of *Oct3/4*, *Sox2*, *Klf4*, *c-Myc*, and *Rex1* were at the same level or higher than in iPS cells, but *Nanog*, *Eras*, and *Esg1* were significantly lower (Figure 2A). The change in the *Nanog* expression level was opposite, suggesting that the mechanism of the metastasis-suppressive effect is completely different between *Nanog*^+^F10 EVs and iPS EVs.

To examine the effectiveness of EVs as a vaccine, a PBS suspension of EVs (5 μg/100 μL PBS) was introduced into the mouse tail vein three times a week for 3 weeks, nine times in total, according to Figure 2B. Subsequently, on Day 0, *Nanog*^+^F10 cells were introduced with 2.5 × 10^5^ cells/250 μL-PBS. Two weeks later (Day 14), the organs were removed, and the number and volume of metastatic colonies were analyzed. In the control without EVs, 200 metastatic colonies were generated in the liver (Figure 2C1). On the other hand, when *Nanog*^+^F10 EVs and iPS EVs were introduced, the numbers were lower, with 92 and 86 colonies, respectively. In addition, the number of metastatic colonies in the lungs was 70, 39, and 21 when PBS (control), Nanog+F10 EVs, and iPS EVs were introduced, respectively (Figure 2C2). In the liver and lungs together, the number of metastatic colonies was halved compared to the controls (Figure 2C3). These results indicated that metastasis-suppressive effect was obtained with either iPS EVs or *Nanog*^+^F10. Statistically, however, there were no significant difference between iPS EVs and *Nanog*^+^F10. 

**Figure 2 ijms-24-17206-f002:**
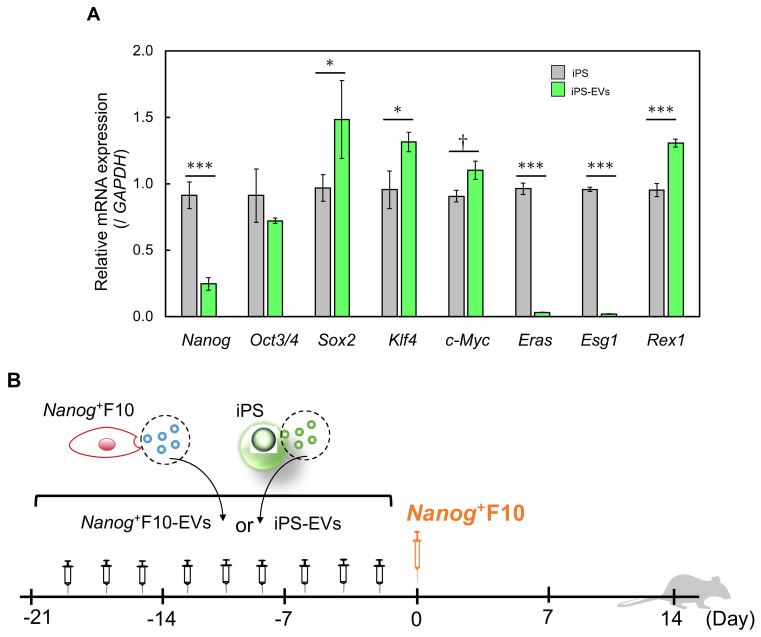
Metastasis-suppressive effects of *Nanog*^+^F10 EVs and iPS EVs in mice. (**A**) Expression of undifferentiated state-associated genes in iPS EVs compared to iPS. mean ± SD, n = 3, † *p* < 0.1, * *p* < 0.05, *** *p* < 0.001. (**B**) Schedule for the administration of EVs and *Nanog*^+^F10 cells. (**C1**) Representative images, number (N), and volume (V) of metastasis colonies in the liver. (**C2**) Same as (**C1**) for lung. (**C3**) Total colonies of the liver and lung. Black arrows in (**C1**,**C2**) indicate examples of metastatic colonies. n = 6 for *Nanog*^+^F10 EVs, n = 7 for others excluding outliers (o). † *p* < 0.1, * *p* < 0.05, ** *p* < 0.01.

### 2.3. Effects on Immune Cells

It was speculated that the vaccine’s effectiveness was achieved through immune cells, and in vitro experiments were conducted using the cultured cell lines of macrophage J774.1 and cytotoxic T-cell CTLL-2. Regarding J774.1 cells, the inhibitory effect on the invasion ability of *Nanog*^+^F10 cells and the phagocytic activity using nanobeads were investigated. Regarding CTLL-2 cells, the cytotoxic activity against *Nanog*^+^F10 cells was investigated.

Regarding the inhibitory effect on invasion ability, we sorted only J774.1 cells loaded with EVs based on the flow cytogram in Figure 3A, and co-cultured them with *Nanog*^+^F10 cells on a Transwell porous membrane coated with Matrigel^®^. Only *Nanog*^+^F10 cells have matrix metalloproteinase (MMP), an enzyme that degrades Matrigel, and can invade Matrigel and pass through the porous membrane. The number of cells that finally permeated the Transwell membrane was calculated. As a result, compared to the control, in the case of *Nanog*^+^F10 EVs, it was reduced to less than 1/10, and in the case of iPS EVs, it was further reduced to 1/2 compared to the *Nanog*^+^F10 EVs (Figure 3B).

Phagocytic activity was analyzed using the nanobead uptake method. J774.1 cells were loaded with PKH26-labeled EVs. Then, yellow–green fluorescent beads (YG-beads) were added to the J774.1 cell suspension. After incubation for 4 h, the fluorescence of J774.1 cells was analyzed using flow cytometry (Figure 3C). Phagocytic activity was calculated from the percentage of cells that incorporated YG-beads. The number of beads taken up by iPS EV-loaded cells increased (27%, *p* < 0.05), but did not increase in *Nanog*^+^F10 EV-loaded cells (Figure 3D).

To analyze cytotoxic activity, EV-loaded CTLL-2 cells were separated using a cell sorter (Figure 3E) and co-cultured for 24 h with *Nanog*^+^F10 cells at a ratio of CTLL-2:*Nanog*^+^F10 = 50:1. As a result, cytotoxic activity (% of dead cells) was calculated from the number of dead cells. Compared to the control, the cytotoxic activity of iPS EV-loaded cells increased by 51% (*p* < 0.001) but did not increase in *Nanog*^+^F10 EV-loaded cells (Figure 3F).

These results indicated that both EVs were effective in enhancing the metastasis-suppressing ability of immune cells. The effects of iPS EVs on immune cells were greater than *Nanog*^+^F10 EVs in all categories.

**Figure 3 ijms-24-17206-f003:**
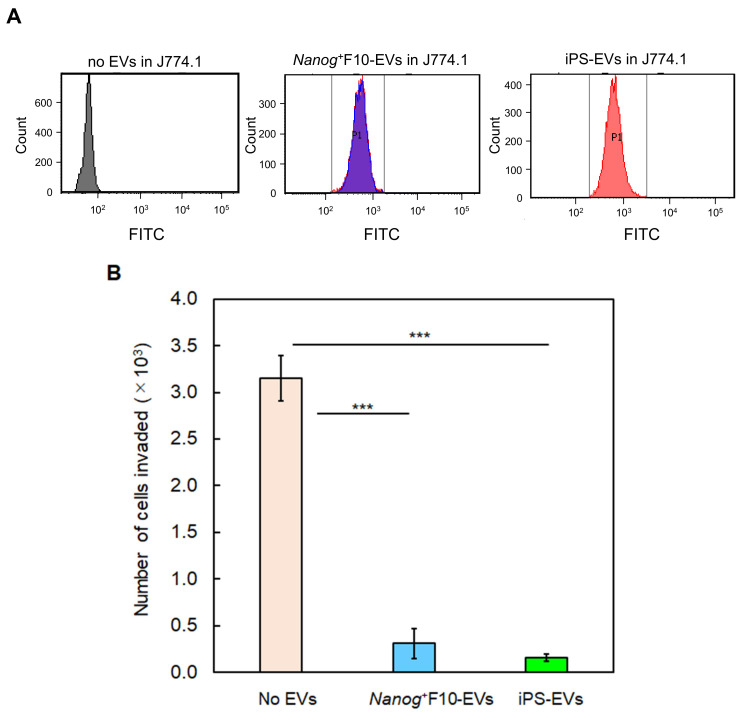
Effects of EVs on the metastasis-related properties in cultured cell lines of J774.1 and CTLL-2. (**A**) Flow cytograms of J774.1 cells loaded with EVs labeled with SYTOTMRNASelectTM (Ex: 490 nm, Em: 530 nm) detected by an FITC detector. a: J774.1 cells containing no EVs, b: *Nanog+*F10 EV-loaded cells, c: iPS EV-loaded cells. The P1 fraction was sorted for the invasion test. (**B**) Number of *Nanog+*F10 cells that invaded Matrigel and passed through the Transwell® membrane. mean ± SD, n = 3, *** *p* < 0.001. (**C**) Flow cytograms of J774.1 cells containing PKH26-labeled EVs and/or YG-beads. PKH26 (Ex: 551 nm, Em: 567 nm) and YG-beads (Ex: 441 nm, Em: 486 nm) were detected using PE and FITC detectors, respectively. (**D**) The number of J774.1 cells that phagocytosed YG-beads. mean ± SD, n = 3, * *p* < 0.05. (**E**) Flow cytograms of CTLL-2 cells loaded with EVs labeled with SYTOTMRNASelectTM. EV-loaded cells mainly existed in the P1 fraction and were sorted for the cytotoxicity test. (**F**) The effects of EVs on the cytotoxicity of CTLL-2 cells against *Nanog+*F10 cells. mean ± SD, n = 3, *** *p* < 0.001.

### 2.4. Results of miRNA Analysis

The differential analyses of miRNA expression levels in metastasis-promoting F10 EVs versus metastasis-suppressive *Nanog*^+^F10 EVs, and in iPS EVs versus *Nanog*^+^F10 EVs were conducted using Macrogen’s Mus musculus Small RNA Sequencing software (ver.MGSR 3.0_mm10). The number of miRNAs whose expression was statistically significantly (*p* < 0.05) increased (Up: Fold change (Fc) > 2) or decreased (Down: Fc < 1/2) was 10 in *Nanog*^+^F10 EVs (Up: 4, Down: 6) (Figure 4A) and 80 in iPS EVs (Up: 37, Down: 43) (Figure 4B). Among the miRNAs with increased expression, we selected up to 16 miRNAs (Table 1) in descending order of Fc, and the target genes of each miRNA were determined using TargetScanMouse_8.0. The selection criterion for miRNAs that were sufficiently knocked down (−25%) with high reliability was CWCS < −0.4. The total number of genes selected from 16 miRNAs was 1074 (Table 1, 5th column). From these target genes, we proceeded with two approaches to predict candidate hub genes that were thought to hold the key to the metastasis suppression mechanism: global data-based (GD-based) analysis and specific keyword-based (SK-based) analysis.

For GD-based analysis, the functional protein association networks of proteins corresponding to 1074 genes were analyzed using STRING ver.12.0, and the results were exported to the network display software Cytoscape ver. 3.10.1. The cytoHubba program, which analyzes association levels, was installed in Cytoscape in advance and applied to the prediction of the top 30 genes that were considered to have high association levels and high importance. We created miRNA-hub gene candidate networks including the top 30 genes and the closely related 11 miRNAs (Table 1, 6th column) (Figure 4C). In the circular layout, it can be seen that among the 30 genes, those distributed on the lower right side show a high degree. Specifically, we selected ten candidate genes with high degrees (Table 2).

On the other hand, in the SK-based analysis, keywords presumed to be related to metastasis inhibition were set in advance for each of three categories (18 keywords in total, Table 3). Among the annotation descriptions of the 1074 genes provided by the social database, we found 175 genes that contained at least one keyword. The presence or absence of keyword(s) in each annotation description was interpreted as the presence or absence of keyword-gene network(s) and their networks between 18 keywords and 175 genes were visualized (Figure 4D). Specifically, we selected six candidate genes with a degree of 4 (Table 4). These genes did not overlap with those listed in Table 2. Therefore, we proposed the 16 genes listed in Table 2 and Table 4 as the candidates for the feedback for future experimental studies using mice and cells.

**Figure 4 ijms-24-17206-f004:**
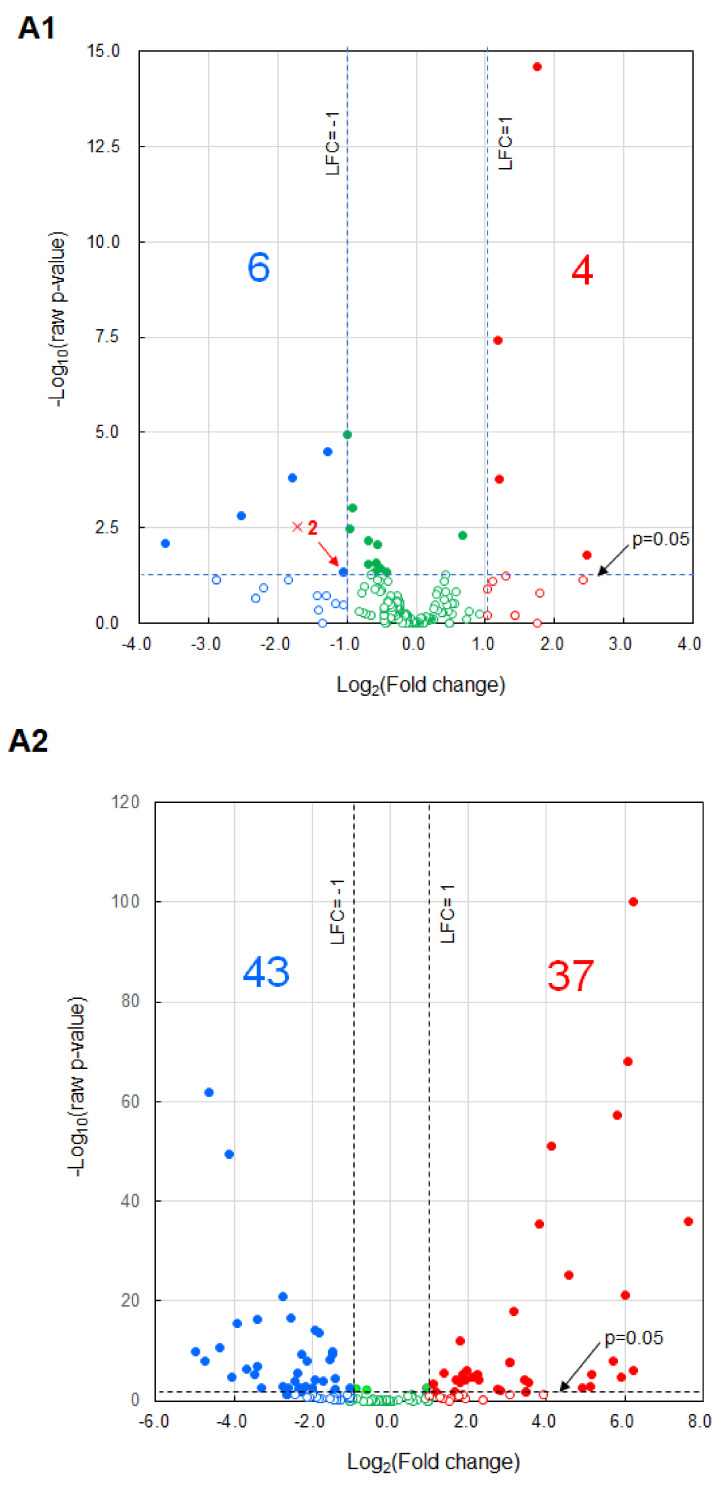
Candidates miRNAs and hub genes associated with the metastasis-suppressive effects. (**A1**) Vol-cano plots of miRNA expression in *Nanog+*F10 EVs. The fold change (Fc) of relative expression magnitude is indicated by Log2(Fc). *p*-values for statistical significance are indicated by −Log10(p). (**A2**) Same as A1 for iPS EVs. “×2” indicates that two miRNAs with same values overlap. Numbers in red are miRNAs that meet the condition of Log2(Fc) > 1 and *p* < 0.05. Numbers in blue are miRNAs that meet the condition of Log2(Fc) < −1 and *p* < 0.05. (**B**) Flow chart of global data-based analysis and specific keyword-based analysis for the prediction of miRNA and gene targets asso-ciated with the metastasis-suppressive effects. (**C1**) Networks between up-regulated and down-regulated miRNAs in *Nanog+*F10 EVs and the most highly associated 30 genes. The network chart was obtained through global data-based analysis. (**C2**) Same as C1 for iPS EVs. All terminals marked * are connected. (**D1**) Networks between 77 genes in *Nanog+*F10 EVs and 18 keywords. Keywords in three categories are listed in Table 3. The network chart was made based on the specific keyword-based analysis. Genes with 4 degrees are listed in Table 4. (**D2**) The networks between 102 genes in iPS EVs and 18 keywords listed in Table 3. Refer to the legend for (**D1**) about other conditions.

## 3. Discussion

This study revealed that iPS EVs showed similar suppressive effects to *Nanog*^+^F10 EVs on melanoma metastasis. Since EVs were introduced in the absence of cancer cells, their effects were thought to be exerted through strengthening of the immune cell system. The analytical stages were (1) to ensure that EVs administered through the tail vein reach the target organ and enter the cells, (2) to clarify which miRNAs in EVs have significantly changed expression, (3) to predict target genes of respective miRNAs based on public big data, and (4) to focus on a small number of genes important for mechanism analysis. At stage (4), we employed two methods: GD-based analysis and SK-based analysis. The SK-based analysis was introduced to enable the focused selection of genes associated with immune responses and inflammation signaling. Finally, we predicted 16 genes as the candidates of hub genes (Table 5). At the same time, it was found that six miRNAs were involved in regulating the expression of these genes.

Hub gene candidates need to be evaluated from two viewpoints. The first viewpoint concerns whether a correlation between the gene expression level and the metastasis suppressive effects (via immune system activation or direct tumor suppression) is positive or negative. In the “Hub gene candidates” column in Table 5, the genes are listed separately in light blue boxes and pale red boxes. Genes in the light blue boxes are required to be negative. This is because the metastasis suppression effect was obtained under the condition that associated miRNAs were up-regulated and, consequently, those genes were down-regulated.

*Trp53*, for instance, is transformation-related protein 53, while wild-type is a tumor suppressor; in contrast missense-type mutant p53 plays a tumor-promoting role [26]. If the gene is wild-type, its ability to suppress tumors decreases as its expression level decreases. Therefore, the correlation can be said to be positive. In contrast, *Hif1a* is a hypoxia-inducible factor that responds to oxygen deprivation and is temporarily protective; however, over a long period it causes pathological reactions in various tissues. For example, prostatic intraepithelial neoplasia is initially hypoxic, and the presence of Hif1a promotes malignant progression [27]. This suggests that Hif1a is also metastasis-promoting rather than metastasis-suppressive. This case may be said to be negative.

The second viewpoint concerns whether the role of a gene is closely relevant to the immune system or inflammation signaling. In the conventional data search method (referred to as GD-based analysis in the text), genes related to cell proliferation appear frequently, regardless of cell type. Therefore, we used a search method that selected genes that were more closely related to immune responses and inflammation signaling, that is, SK-based analysis. In the “Predicted by” column of Table 5, these two methods are indicated by green and yellow boxes, respectively. For instance, *Kitl*, *Fgf16*, *Grb2*, and *Fgf1* were selected through the GD-based analysis. In fact, although these genes are involved in controlling cell proliferation, they do not seem to be particularly relevant to the immune system.

However, despite the above discussion, it is necessary to conduct validation experiments for all predicted genes. This is because there are no reports yet regarding their effects on the specific target of melanoma metastasis. Next, genes that have been proven to be effective will be further validated through in vivo mouse experiments using, for example, lipid nanoparticles.

## 4. Material and Methods

### 4.1. Cell Culture

Mouse melanoma cell lines used in this study were wild B16-F10 (F10) and *Nanog*-overexpressing F10 (*Nanog*^+^F10). These cells were cultured under the conditions described in [18]. The mouse induced pluripotent stem (iPS) cell line was iPS-MEF-Ng-20D-17 and was purchased from RIKEN BRC. iPS cells were cultured in iPS medium on a feeder cell layer of mouse embryonic fibroblasts (MEFs) or in a 0.1% gelatin-coated culture dish. The iPS medium per 50 mL contained 41.4 mL of DMEM (high glucose, no sodium pyruvate), 7.5 mL of fetal bovine serum, 0.5 mL of NEAA, 0.05 mL of 2-mercaptoethanol, 0.05 mL of mouse LIF, and 0.5 mL penicillin–streptomycin. Culturing was conducted at 37 °C under 5% CO_2_. iPS cells cultured on feeder cells were replaced with medium every day and passaged every three days. Passaging was conducted using treatment with 0.25% trypsin-EDTA for 2 min and centrifugation at 1000 rpm and 20 °C for 5 min. Mouse macrophage J774.1 and mouse cytotoxic T-cell line CTTL-2 were obtained from RIKEN BRC. J774.1 cells were cultured in the same manner as mouse melanoma cells [18]. CTLL-2 cells were cultured in suspension in a medium containing 50 unit/mL mouse interleukin-2 (IL-2) (WAKO).

### 4.2. Preparation of EVs

EVs were separated according to [18]. Cells were cultured in PMI1640 (containing 10% EV-depleted FBS and 1% penicillin–streptomycin) for 48 h. After centrifugation at 2000× *g* for 20 min, the supernatant was filtered through a 0.22 μm pore size and centrifugated at 100,000× *g* for 80 min. The mode diameter and mean diameter of EVs were determined as approximately 90 nm and 130 nm, respectively, using the NanoSight NS300 system (Malvern Panalytical, Malvern, UK). The EV fraction was confirmed using a CD81 marker.

### 4.3. Western Analysis of CD81 and Gapdh

A protein sample of cells was prepared according to the following procedure. After rinsing 70–80% confluent cells with PBS, an RIPA buffer (25 mM Tris-HCl, 150 mM NaCl, 1% NP-40, 1% sodium deoxycholate, 0.1% SDS, pH 7.6; Thermo Fisher Scientific, Waltham, MA, USA) was added to the culture dish. The dish was stood on ice for 15 min. Then, the cells were peeled from the culture dish with a cell scraper and collected in a 1.5 mL microtube. The cell suspension was sonicated (UR-20P; TOMY SEIKO, Tokyo, Japan) on ice and centrifugated at 20,000× *g* for 15 min. The supernatant was collected as a protein sample of cells. A protein solution of EVs was prepared as follows. The pellet of EVs obtained as the precipitate of ultracentrifugation was suspended in an RIPA buffer and stood on ice for 15 min. The protein concentration was determined using a Pierce^®^ BCA TM Protein Assay kit (Thermo Fisher Scientific). A protein solution was mixed with a 1/6 volume of 0.375 M Tris-HCl (pH 6.8) buffer solution containing 93 μg/mL DTT, 0.12 g/mL SDS, 0.6 mL/mL glycerol, and 0.6 mL/mL bromophenol blue. Then, the solution was heated at 95 °C for 5 min and applied to SDS-PAGE at 150 V. Blotting onto a PVDF membrane was conducted at 100 V for 3 h at 4 °C. The PVDF membrane was then immersed in a TBS-T (Tris-buffered saline (25 mM Tris, pH 7.4, 150 mM NaCl) containing 1 (*v*/*v*) % Tween 20) solution containing 5% (*w*/*v*) skim milk at 25 °C for 30 min. Then, the PVDF membrane was incubated in a 5% skim milk TBS-T solution containing primary antibody at 25 °C for 3 h. Primary antibodies against mouse Gapdh (1:1000, sc-32233; Santa Cruz Biotechnology, Dallas, TX, USA) and mouse CD81 (1:500, sc-166029; Santa Cruz Biotechnology, Dallas, TX, USA) were used, respectively. After washing with TBS-T three times, the membrane was incubated in TBS-T containing a secondary antibody (anti-mouse immunoglobulin conjugated to alkaline phosphatase, Promega, Madison, WI, USA) at 25 °C for 1 h. Membranes were washed three times with TBS-T and then incubated with Western Blue Stabilized Substrate (Promega) for alkaline phosphatase at 25 °C for 5 min.

### 4.4. Analysis of the Transfer of EVs to Various Organs in a Mouse

EVs were stained with CellVueTM NIR815 (Invitrogen) and introduced into mice through tail vein injection at 5 µg/100 μL PBS. The mice were anesthetized at predetermined time intervals and their whole bodies were imaged using a Pearl Trilogy imaging system (LI-COR) at 800 nm. At the same time, various organs were extracted from another group of mice for imaging analysis. Based on the results, the optimum time for the extraction of organs was determined at 3 h after introduction, since the amount of EVs in each organ reached its maximum or steady state around 3 h. The integrated fluorescence intensity of the entire organ was divided by the organ area (signal/area) to estimate the density of EVs accumulated in each organ.

### 4.5. In Vivo Test of the Effects of EVs on the Metastasis of Melanoma

Experiments were conducted according to the protocols described previously [18]. Briefly, EVs were introduced repeatedly through tail vein injections of 5 µg/100 μL-PBS. Then, *Nanog*^+^F10 cells were introduced through tail vein injections of 2.5 × 10^5^ cells/250 µL-PBS. After two weeks, livers and lungs were removed, and the number and the volume of each metastatic colony were analyzed per mouse.

### 4.6. Invasion Ability Test

EVs were stained with CYTO RNASelect (Ex: 490 nm, Em: 530 nm) (Invitrogen) and introduced into J774.1 cells. Only cells that had taken up EVs were sorted using a cell sorter (FACSAriaII, BD, Franklin Lakes, NJ, USA) and co-cultured with *Nanog*^+^F10 cells on Matrigel that was coated on a porous membrane of a Transwell^®^ 24-well plate (Corning, Corning, NY, USA). During culturing for 22 h, *Nanog*^+^F10 cells alone could generate matrix metalloproteinase (MMP) to hydrolyze and then invade Matrigel. The number of cells that passed through Matrigel and the porous membrane was analyzed as follows: Twenty small rectangular areas were selected and the number of cells observed within each area was counted and summed up. The total number of cells that passed through whole area of the membrane was estimated from the data obtained from the small areas.

### 4.7. Phagocytic Activity Test

The phagocytic activity of J774.1 cells was estimated using the microbead uptake test. EVs were stained with a fluorescent dye, PKH26 (Ex: 551 nm, Em: 567 nm, DOJINDO, Kumamoto, Japan), and introduced into J774.1 cells. Yellow–green fluorescent beads (YG-beads) (Ex: 441 nm, Em: 486 nm, Sigma, St. Louis, MO, USA) were added to a culture dish of J774.1 cells. After culturing for a certain period of time, cells were analyzed using FACSAriaII.

### 4.8. Cytotoxic Activity Test

EVs were stained with CYTO RNASelect and introduced into CTLL-2 cells. Only cells that had taken up EVs were sorted using FACSAriaII. The cytotoxic activity of EV-loaded CTLL-2 cells was tested using an LDH cytotoxic assay kit (Dojindo). CTLL-2 cells were co-cultured with *Nanog*^+^F10 cells in 100 μL EV-depleted RPMI medium. During culturing for 24 h, lactate dehydrogenase (LDH) leaked into solution from dead cells. After the culture, a working solution containing lactate, NAD^+^, and water-soluble tetrazolium (WST) was added to advance the LDH reaction. Consequently, WST was reduced to WST-formazan that was colored and quantified through absorbance at 490 nm. A high-control solution was prepared by adding a cell lysis buffer to a sample cell suspension to obtain the maximum LDH release from the sample cells when all the cells were killed. A low-control solution was a sample cell suspension without any treatment to obtain LDH due to spontaneous release from cells. Cytotoxic activity was determined as follows:

Cytotoxicity (%) = {[OD490 of a test sample solution] − [OD490 of the low control solution]}/{[OD490 of the high control solution] − [OD490 of the low control solution]}×100.

### 4.9. miRNA Analysis

*Nanog*^+^F10 EVs and iPS EVs were metastasis-suppressive, but, contrarily, F10 EVs were metastasis-promoting. To predict gene candidates that are closely associated with the metastasis-suppressive effect, a differential expression analysis of miRNA expression in respective EVs was performed. RNA extraction from EVs and small RNA sequencing analysis (ver. MGSR3.0_mm10) were outsourced to Macrogen Japan (Koto, Tokyo, Japan). The magnitude of variation and the statistical significance (*p* value) of respective miRNA components were obtained.

### 4.10. Statistical Analysis

Specific details regarding statistical analyses are presented in the figure legends. Each test sample was analyzed twice or three times, and the average of the two or three results was recorded as the value for one test sample. The results were presented as the mean and standard deviation (SD) for the number of samples (n). The results of the metastatic colony analyses were presented in box plots. Outliers shown in box plots were determined by a Smirnoff–Grubbs test to be greater than 0.05 one-tailed probability. The statistical significance between two specific data groups was analyzed using a two-tailed Student’s *t* test. The statistical significance of results was denoted by a *p* value or by marking with asterisk(s): ***: *p* < 0.001, **: *p* < 0.01, *: *p* < 0.05, †: *p* < 0.1.

## Figures and Tables

**Figure 1 ijms-24-17206-f001:**
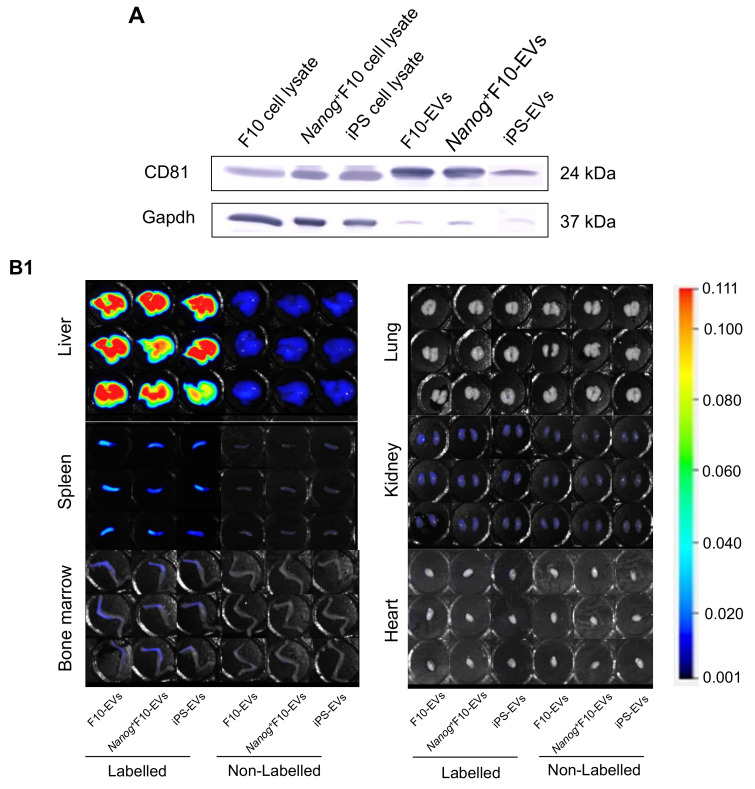
Accumulation of EVs in eight organs. (**A**) Confirmation of CD81-positive fractions of EVs. (**B1**) Imaging of F10 EVs, *Nanog+*F10-EVs, and iPS EVs in liver, lung, spleen, kidney, bone marrow, and heart. EVs were labeled with CellVue™ NIR815 or unlabeled. n = 3 mice per each of six con-ditions. (**B2**) Same as (**B1**) for stomach and intestine. (**C**) Comparison of the accumulated densities of EVs in different organs. Mean ± SD, n = 3, ** *p* < 0.01.

**Table 1 ijms-24-17206-t001:** Summary of up-regulated and down-regulated miRNAs in *Nanog*^+^F10 EVs and in iPS EVs; total number of predicted target genes with high efficiency; top 30 genes with the highest association among them.

	miRNA	Log_2_(FC)	Number of Target Genes		
			*p* < 0.05CWCS ≤ −0.4		Targeted by Each miRNA among Top30 Genes
*Nanog*^+^F10-EVs/F10-EVs	miR-18a-5p	2.46	42	301	4
	miR-3473e	1.75	6		0
	miR-19a-3p	1.20	126		11
	miR-3473b	1.18	127		5
	miR-210-3p	−3.62	6	104	0
	miR-369-3p	−2.52	7		2
	miR-122-5p	−1.79	29		5
	miR-706	−1.28	62		3
			Sum	405	30
iPS-EVs/*Nanog*^+^F10-EVs	miR-466h-3p	7.62	58	275	1
	miR-706	6.23	62		3
	miR-323-3p	6.23	10		0
	miR-466f-3p	6.07	145		5
	miR-30a-3p	−4.99	49	394	0
	miR-342-5p	−4.74	335		20
	let-7c-5p	−4.63	2		0
	miR-423-3p	−4.36	8		1
			Sum	669	30

**Table 2 ijms-24-17206-t002:** Candidates of hub genes and associated miRNAs summarized from Figure 4C.

	Gene	MCC Score	Degree	Regulation by mi RNA
*Nanog*^+^F10-EVs	Trp53	1058	23	up-regulated-miR-3473b
	Hif1a	990	19	up-regulated-miR-18a-5p
	Esr1	919	11	up-regulated-miR-18a-5p
	Atm	872	9	up-regulated-miR-18a-5p
	Cdkn1b	847	7	down-regulated-miR-706
iPS-EVs	Ins1	13,801	19	up-regulated-miR-466f-3p
	Kitl	13,493	15	up-regulated-miR-466f-3p
	Fgf16	13,254	11	up-regulated-miR-466f-3p
	Grb2	11,784	15	up-regulated-miR-466f-3p
	Fgf1	11,640	11	down-regulated-miR-342-5p

MCC: maximal clique centrality.

**Table 3 ijms-24-17206-t003:** Keywords determined in this study for the specific keyword-based analysis.

Specific Keywords			
**Immune Responses**	**Immune Cells**	**Inflammation Signaling**	
Immune	T cell	Inflammation	JAK/Stat
Cytotoxicity	B cell	Cytokine	CD3
Phagocytosis	Dendritic	Interleukin	CD4
	Helper	NF-kappaB	CD8
	Macrophage	TGF-beta	
	Myeloid		

**Table 4 ijms-24-17206-t004:** Candidates of hub genes and associated miRNAs summarized from Figure 4D.

	Gene with 4 Degrees	Regulation by miRNA	Associated Keywords
*Nanog*^+^F10-EVs	Rnf11	up-regulated-miR-19a-3p	Immune, Cytokine, NF-kappaB, Tgf-beta
iPS-EVs	Malt1	up-regulated-miR-466f-3p	Immune, T cell, B cell, Helper cell
	Il1f6	down-regulated-miR-30a-3p	T cell, Dendritic cell, Interleukin, NF-kappaB
	Cmklr1	down-regulated-miR-342-5p	Immune, Macrophage, Inflammation, NF-kappaB
	Siglec1	down-regulated-miR-342-5p	T cell, Macrophage, B cell, CD8
	Cd28	down-regulated-miR-342-5p	Cytotoxicity, Immune, T cell, CD4

**Table 5 ijms-24-17206-t005:** Summary of predicted hub genes and miRNAs.

EVs	miRNA	Target Genes	Prediction By	Hub Gene Candidates
*Nanog*^+^F10-EVs/F10- EVs	up-regulated miRNAs (miR-18a-5p, miR-19a-3p, miR-3473b)	down-regulation of target genes	GD-based analysis	Trp53
				Hif1a
				Esr1
				Atm
			SK-based analysis	Rnf11
	down-regulated miRNAs (miR-706)	up-regulation of target genes	GD-based analysis	Cdkn1b
			SK-based analysis	-
iPS-EVs/*Nanog*^+^F10-EVs	up-regulated miRNAs (miR-466f-3p)	down-regulation of target genes	GD-based analysis	Ins1
				Kitl
				Fgf16
				Grb2
			SK-based analysis	Malt1
	down-regulated miRNAs (miR-342-5p)	up-regulation of target genes	GD-based analysis	Fgf1
			SK-based analysis	Il1f6
				Cmklr1
				Siglec1
				Cd28

## Data Availability

The datasets generated and/or analyzed during the current study are available from the corresponding author on reasonable request.

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
