# Peer review of "Comparative Study of Metastasis Suppression Effects of Extracellular Vesicles Derived from Anaplastic Cell Lines, Nanog-Overexpressing Melanoma, and Induced Pluripotent Stem Cells"

_ijms, 2023, doi:10.3390/ijms242417206_

Round 1
Reviewer 1 Report
Comments and Suggestions for Authors
Extracellular vesicles (EVs) are known to act as shuttles to deliver functional materials to neighboring cells. The authors previously demonstrated that the EVs from Nanog-overexpressing cells (Nanog+F10-EVs) exert an unexpected metastasis suppressive effect. In this follow-up study, the authors compared the Nanog+F10-EVs to the EVs from an induced pluripotent stem cell line (iPS-EVs). The anti-metastatic potential in vivo was assessed by analyzing colonies in lung and liver tissues after tail-vein injection. Immune modulation by EVs was examined based on phagocytotic and cytotoxic activities. Comparative analysis of miRNAs and target genes between Nanog+F10-EVs and iPS-EVs was performed. The main issue with this study is that the logic of comparing EVs from two very different sources is dubious. The in vivo anti-metastatic potential of Nanog+F10-EVs and iPS-EVS should be compared to wild-type F10-EVs rather than no EVs (or PBS). The vaccinating effect of EVs should be tested using mouse-derived peripheral blood dendritic cells and T cells, not established cell lines, such as J774.1 and CTLL-2. Although miRNA analysis and target prediction identified several candidate hub genes associated with anti-metastatic effects, it is unclear whether these results are meaningful because they have not been validated.
1. In Figure 1A, what does the red box mean? Nanog mRNA level is lower in iPS-EVs than original iPS cells, but other stem cell marker genes are higher. Could these genes have anti-metastatic potential? In Fig. 1C and D, it would be better to indicate tumor nodules within the tissues using solid arrows. In Figure 1 legend, please indicate what N and V mean.
2. In Figure 2A, please put labels above the histograms. I guess that the values in Fig. 2D were from the P1 population (double positive) in Fig. 2C. But, in the PBS group, the P1 is almost empty, but Fig. 2D shows many cells phagocytosed particles at 4 h.
3. Differentially expressed miRNAs were identified in Nanog+F10-EVs and iPS-EVs in comparison to wild-type F10-EVs. The comparison of wild-type F10-EVs to Nanog+F10-EVs looks good, but not to iPS-EVs. It might be better to compare iPS-EVs to Nanog+F10-EVs directly.
4. In Tables 2 and 4, the authors proposed 12 candidate genes associated with anti-metastatic activity, but did not evaluate whether these genes were actually downregulated in the iPS-EVs. If overexpression of some of the candidate genes abrogates the antimetastatic activity of iPS-EVs, it would strengthen their conclusions.
5. The discussion section needs to be re-written.
Comments on the Quality of English Language
No issues.
Reviewer 2 Report
Comments and Suggestions for Authors
The authors provide interesting data in this manuscript. However, the manuscript still needs some improvements before publication eligibility.
- Figure 1 A Size of bands needs to be added
- Figure 1 B x axis legend is needed , in image and not just legend. Groups should be added. If the photos from 4-6 are not labelled so what does the blue stain signify ?
- In figure 1 data, n=3 is quite low and Nanog+F10-EVs and iPS-EVs show high error bars in the liver samples.
- Line 113, I think something is wrong with these numbers "1,3: F10-EVs. 2,4: Nanog+F10-EVs. 3,5: iPS-EVs".
- Line 117 -118, how come we are comparing these 2 models despite that the nanog expression is complete opposite between them. Is the comparison based on metastatic suppression potential ?
- Flow cyometry graphs need to be organized and labelled. Also the representative images need to include percentages of each cell population.
- Line 199 (Down: Fc<1/2) is confusing. Better to mention that log Fc is 1 and -1 to be matching with the figures.
- Line 201, 5 types of miRNAs , what do you mean by types?
- Line 204, miRNAs should be mRNAs or genes
- Why were only upregulated miRNAs used for further analysis ?
- It would have been interesting if the authors were able to identify pathways shared by the identified genes if they are associated with metastasis suppression as well
- Discussion needs major rewriting to further highlight the importance of presented results with previous literature and not just describe the role and functions of identified genes
- Abbreviations need to be mentioned at their first mention.
Reviewer 3 Report
Comments and Suggestions for Authors
Here the authors present the overexpression of an embryonic marker, Nanog, on melanoma cell line B16F10 (Nanog+F10). They isolated EVs secreted by Nanog+F10 and compared them to secreted EVs by iPS cell line.
Some concerns should be addressed before being considered for publication.
Major comments:
Ideas present on literature revision are confusing, e.g., the first paragraph should be rewritten. There is no definition of extracellular vesicles, which are they? What are their importance in pathophysiology? What is evidence supporting EVs acting as effective metastasis-inhibiting vaccines? They referred to a review that does not discuss it (ref 6). There is missing a reference for the statement on line 39. In 3rd paragraph, most references showing the importance of Nanog expression on melanoma are self-citations. There is no further relevant literature on the role of Nanog on melanoma cells.
The manuscript presents rational design inconsistencies. They showed accumulation of EVs only in the liver and no minimal signal in the lungs, however, they evaluated experimental metastasis in both organs (Figure 1 shows no accumulation of EVs in lungs, however Figure 2 shows lung metastasis after EVs pre-treatment, how is it EVs dependent?).
Figure 2A shows the expression of different anaplastic genes on iPS cells compared to iPS-EVs. What about Nanog+F10-EVS expression of the same genes? Was it supposed to upregulate these genes when compared to parental cells, B16F10?
Next, the authors pre-treated animals with EVs isolated from Nanog+F10 and from iPS cells, in different concentrations used before in Figure 1 (see line 89 and line 125). Then, the authors injected B16F10 cells into the tail vein and evaluated the number of colonies in the liver and in the lung. It is missing a group control (EVs parental cells, B16F10). Additionally, the number of colonies observed in the PBS control group is much higher than in previous data published by authors (Hatakenaka T., 2022), even using the same methods. As the number of colonies presented here is too high (around 200 colonies per liver), authors should confirm that data performing an HE histology. The organs (livers and lungs) presented in Figure D are too red due to excess of blood. It is not convincing that authors count metastasis focus or even clots. Moreover, the tumor volume of each organ should be corrected by the weight of each tumor-bearing mouse, as they may present weight losses. Finally, authors should be careful of overstatements, since data present in these figures do not show statistical differences between Nanog+F10-EVs and iPS-EVs groups.
There are discrepancies in invasion description between the abstract and Figure 2; are macrophages or melanoma cells migrating? On iPS-EVs groups, authors described a migration reduction of half, but compared to which group, it is not clear.
Figure 3C shows the phagocytic activity of macrophage cells upon EV phagocytosis. The authors say a tendency of difference for phagocytic activity, again, please you should be careful with overstatements, there are no statistical differences between the control and Nanog+F10 group (same as figure 3F). Moreover, it is not clear how this cytotoxicity design works: which is a low-control solution, or a high-control solution?
Figure 4 – There is no detailed description of how authors isolated miRNA in the methodology section. Which endogenous control did authors use to normalize miRNA expression? The description of the evaluation of the analysis of selected miRNAs and their target genes is really confusing. First, authors selected the up and downregulated miRNA expressed in Nanog+F10-EVs and iPS-EVs, and they found 10 and 72 miRNAs differently expressed, respectively. For the miRNA-hub gene, they included the top 9 miRNAs, but in Figure 4D we can see only 7 miRNAs. However, in summary, they classified the dominant regulatory miRNA for Nanog+F10-EVs and iPS-EVs but the authors do not discuss these data throughout the manuscript. Finally, they selected 6 target genes and described their original function in the discussion section. It should be more speculative about EVs alteration in metastasis processes.
Minor comments:
EVs isolation methodology is barely described, how did authors measure EVs diameter, for instance? How did they evaluate CD81 and GAPDH expression?
Please, it should perform a comprehensive review for formatting and language improvements throughout the manuscript.
Comments on the Quality of English LanguageAuthors should review extensively the quality of the English language using more scientific terms. It is not appropriate to write "cancer cells are killed by" or "cancer cells that make up cancer tissue" or even "every item relevant to metastasis".
Round 2
Reviewer 1 Report
Comments and Suggestions for Authors
The authors addressed most of the reviewer's concerns, but the English still needs work.
Comments on the Quality of English LanguageThere are some errors in English grammar that need to be reviewed by a native speaker.
Author Response
Thank you for comment. According to the comments, we have thoroughly revised the manuscript. We appreciate your kind re-evaluation.
[General 1]
The authors addressed most of the reviewer's concerns, but the English still needs work.
[Res G1]
We have thoroughly checked and revised the manuscript.
Reviewer 2 Report
Comments and Suggestions for Authors
The authors addressed all my previous comments.
Author Response
Thank you for your re-evaluation.
Reviewer 3 Report
Comments and Suggestions for Authors
Here the authors presented a few improvements throughout the text previously pointed out. Some arguments presented by the authors should not convince the manuscript reader, it is still missing proper arguments in the text. Besides, some concerns were not clearly discussed. Some examples are as follows.
The introduction section was rewritten but the first paragraph (more than 12 lines) has basically only 2 references. There is no reference in Line 41 supporting the evidence stated in the text: “EVs derived from immune cells exhibit tumor suppressive effects”. As a reviewer, I could not see enough evidence supporting the use of EVs as a promisor candidate to improve the vaccine technology. I am happy that the authors are positive about the future scope of EVs vaccine technology. However, it is important to clearly state that evidence, always bringing up the current literature.
The rational design of the experiments still presents inconsistencies. For example, the authors showed EVs biodistribution in a mouse, and they showed that EVs accumulate mainly in the liver. Next, they evaluated lung metastasis and did not point out this rationale clearly as EVs do not accumulate in the lung. Regarding lung and liver met colonies, the number varies from 80-380 “depending on time of year”. It is not comprehensive; this observation should be discussed based on literature data. Moreover, the pictures presented in the manuscript do not reflect even close to the number of counted colonies, i.e., it does not convince. Microscopically, the authors could estimate the tumoral area percentage and consolidate their data. Some concerns raised before are not clear in the reviewed manuscript (the particle analyzer, Nanosight; what are low and high control solutions in LDH assay). Moreover, the authors say that they corrected some minor points, but the reviewed text remains unaltered (Res 4-1; Res 6; Res 8).
The authors overstated in the first version of the manuscript the differences between EV_iPS and Nanog-F10_EVs groups. The reviewed manuscript reports no differences between these groups. However, the authors keep stating that EVs_iPS does present higher suppressive metastatic effects than Nanog-F10_EVs, which additionally raises dubious effects of Nanog relevance.
Comments on the Quality of English LanguageAuthors should review extensively the quality of the English language, mainly in the Results section.
Author Response
Thank you for valuable and encouraging comments. According to the comments, we have thoroughly revised the manuscript. We appreciate your kind re-evaluation.

Round 3
Reviewer 3 Report
Comments and Suggestions for Authors
The authors presented some improvements however, the text is still confusing and does not convince the manuscript reader. The article has serious flaws and the research does not seem to be conducted correctly. The data presented here are not enough to evidence that EVs_iPS and Nanog-F10_EVs act as suppressive immune agents and the data raise doubts about Nanog's importance in that context. Said that, I do not recommend accepting this paper.
Comments on the Quality of English LanguageExtensive editing of English language should be required.